# Green Exercise: Can Nature Video Benefit Isometric Exercise?

**DOI:** 10.3390/ijerph18115554

**Published:** 2021-05-22

**Authors:** Hansen Li, Xing Zhang, Shilin Bi, Haowei Liu, Yang Cao, Guodong Zhang

**Affiliations:** 1Key Lab of Physical Fitness Evaluation and Motor Function Monitoring of General Administration of Sports of China, Institute of Sports Science, College of Physical Education, Southwest University, Chongqing 400715, China; hanson-swu@foxmail.com (H.L.); haoweiliu@email.swu.edu.cn (H.L.); 2Department of Basketball and Volleyball, Chengdu Sport University, Chengdu 610041, China; starz-94@foxmail.com; 3National Institute of Education, Nanyang Technological University, Singapore 637616, Singapore; NIE20.BS@E.NTU.EDU.SG; 4Clinical Epidemiology and Biostatistics, School of Medical Sciences, Örebro University, 70182 Örebro, Sweden; 5Unit of Integrative Epidemiology, Institute of Environmental Medicine, Karolinska Institutet, 17177 Stockholm, Sweden

**Keywords:** green exercise, nature, nature-based stimuli, video, indoor exercise, isometric exercise, wall squat

## Abstract

Green exercise is the combination of physical activity and nature exposure, which has been associated with positive effects on psychophysiological health. This study aimed to investigate the effects of nature video viewing on isometric exercise and find a useful practice for green exercise in urban living. In the current study, 18 male subjects were recruited in a randomized crossover trial and underwent a sequence of wall squat exercises. The whole experiment contained three periods of baseline (before exercise), exercise, and recovery (after exercise), and each period lasted for 2 min. A video of forest walking was played in the exercise and recovery periods as treatment, while a black screen was set as control. The Rate of Perceived Exertion Scale (RPE) and Feeling Scale (FS) were employed to measure perceived exertion and affective responses in the exercise period; heart rate (HR) and heart rate variability (HRV) including the standard deviation of normal-to-normal RR intervals (SDNN), the root mean square of successive differences (RMSSD), and the standard deviations of the Poincaré plot (SD1), were recorded in the three periods. Heart rate recovery (HRR) in the recovery period was further calculated based on 30 s and 60 s time frames. Results demonstrated that during the exercise period nature video viewing was associated with better affective responses (median of 1.00 and an interquartile (IQR) of 2.00, *p* = 0.017), lower perceived exertion (median = 6.00, IQR = 2.00, *p* = 0.021), and lower HR (median = 89.60, IQR = 20.94, *p* = 0.01), but the differences in HRV indices between the experimental settings were not statistically significant. In the recovery period, significantly higher values of RMSSD (median = 34.88, IQR = 24.52, *p* = 0.004), SD1 (median = 24.75, IQR = 17.41, *p* = 0.003), and HR (median = 84.18, IQR = 16.58, *p* = 0.001) were observed in the treatment setting, whereas no statistically significant difference was found for HRR. In general, our findings support that nature video viewing may help reduce perceived exertion, increase exercise pleasure, buffer heart rate, and improve cardiac autonomic recovery for wall squat exercising, which implies the potential of nature-based stimuli in green exercise. However, due to the limited research sample, further study may need to include female participants and focus on various populations to confirm the effectiveness of using virtual and environments depicting nature at home or in public exercise places to promote positive exercise experience.

## 1. Introduction

Emerging evidence shows that nature exposure can prevent and treat diseases that result from urban stressors, including air pollution, noise, and crowding [1,2,3]. Therefore, urban dwellers today are encouraged to visit urban green spaces such as urban forests and parks to improve their health conditions [4,5]. However, though urban greening has been advocated and promoted in recent decades, there is still a considerable part of the population who live in highly urbanized areas and have limited access to nature. On the other hand, the benefits of nature are reported to be associated with the quality and characteristics of the natural environment [6], which implies a difficulty in experiencing nature exposure in urban environments. In response, many attempts have been made to deliver the impacts of nature via digital media, including images, audios, videos, and virtual reality [7,8]. These nature-based digital stimuli are found to reduce mental stress, improve cardiac responses, and alter brain activity [8,9,10], which implies the use of nature-based stimuli to improve public health [11,12].

Similar to nature exposure, physical activity is known to reduce chronic conditions caused by urban living, such as type II diseases and mental disorders [13,14]. To achieve a synergetic effect of physical activity and nature, the concept of green exercise (GE) is proposed and illustrated through a series of physical activities in natural environments [15]. As the combination of nature exposure and physical activity may produce additive or sub-additive effects [16], green exercise is deemed to create more health benefits for the public [17,18]. However, despite many studies of green exercise, most focused on the functions of real nature. In contrast, only a few trials were made on the combination of nature-based digital stimuli and physical exercise. According to the published literature, nature-based digital stimuli may decrease perceived exertion and improve recovery from mental stress during exercise [19,20]. However, due to insufficient concern, the other effects of green exercise using nature-based digital stimuli still need exploration. Moreover, previous studies on green exercise have mainly concentrated on green walking, while the effects of other forms of physical activity are yet to be investigated [21,22,23].

Isometric exercise, referring to isometric muscle contraction, has been proved effective in reducing blood pressure and enhancing muscular strength [24,25]. Among various isometric practices, wall squat is the most common exercise that has been applied in athletic training and rehabilitation training [26]. On the one hand, wall squat is a static exercise whose exercise intensity is adjustable according to knee joint angle [27], and is safe for people of different genders and ages [1]. On the other hand, wall squat is an equipment-free exercise and can be carried out anytime under a roof [26], providing an ideal option for public daily exercise. Considering successful cases of green walking, the combination of wall squat and nature exposure may be a beneficial isometric exercise in rehabilitation institutions, households, and indoor sports venues [28,29]. So far, green isometric exercise has not been investigated, and whether nature-based digital stimuli can promote isometric exercise is still a valuable question.

The cardiac response associated with physical activity is a common research interest [30], and heart rate variability (HRV) analysis, a noninvasive method of evaluation based on successive R-R intervals, is useful in indicating sympathetic and parasympathetic response in physical activities [31,32]. Previous studies have detected reduction in HRV during wall squat exercise [33,34], consistent with the phenomenon of most exercises [35]. As yet, many studies indicate that nature-based interventions may help maintain a higher level of HRV during walking exercise [21,22,36], which may be related to better affective states in natural environments. Similar to self-paced walking, wall squat with a larger knee joint angle can be a less-strenuous exercise that may result in less physiological pressure, thus may promote better environmental effects. However, the effectiveness of nature-based interventions on isometric exercise is still unclear, and further investigation is needed.

Nowadays, digital screens are widely equipped in households and public areas, including hospitals, gyms, and rehabilitation centers, and video is the most popular and handy digital media for the public. Considering the known emotional and cardiac benefits of nature [36,37], we hypothesize that nature video may:(1)reduce perceived exertion and improve affective responses during wall squat exercise;(2)buffer heart rate and heart rate variability during wall squat exercise.(3)improve cardiac autonomic recovery after wall squat exercise.

## 2. Materials and Methods

### 2.1. Study Design

A randomized and controlled crossover trial was deployed in the current experiment to investigate the effects of viewing nature-based video on isometric exercise (Figure 1). A nature video was used as the treatment (T) whilst a black screen was used as the control (C). Subjects were randomly divided into two groups (group A and B) and underwent a series of tests under the two experimental settings, in the orders of TCTC or CTCT. Each test contained three periods, P1, P2, and P3, indicating baseline period, exercise period, and recovery period, respectively. 

### 2.2. Participants

The sample size was determined using the statistical software PASS (version 15.0.5, NCSS, LLC. Kaysville, UT, USA). Based on the crossover design, the minimum number of subjects required was 10 at an α-level of 0.05 and with a power of 0.80. As the recruited women had poor connection to the heart rate detector due to their clothing in the initial equipment test, we included 18 young males from Sichuan Agricultural University who met the following inclusion criteria in the experiment:(1)absence of cardiovascular diseases.(2)absence of osteoarthrosis leading to risk during exercise.

The average age of the participants was 27.94 years (Table 1), and all practiced exercise habits (13 basketball and five daily walking), but none was an athlete. The study protocol was approved by the Institute Research Ethics Committee (IREC) of Southwest University, China, conducted in accordance with the Helsinki Declaration, and supervised by the IREC. The signed consent form for participating in the experiment was obtained from each participant before the experiment. All subjects were informed that they could quit at any moment during the experiment at their will. 

### 2.3. Nature Video

A forest walking video was used in the experiment, which displayed a view of walking through a needle leaf and broadleaf mixed natural forest on a sunny day (Figure 2). The video also includes soundtrack (continuous birdsong). The whole video lasted for about 10 min and was played from the beginning for each participant. No video clip was repeated during a single test. The video was displayed using a monitor with a size of 88 cm × 50 cm and a resolution 1080 × 1920, and the audio was delivered using a stereo system (Samsung HW-Q60T).

### 2.4. Wall Squat Exercise

In practical exercise, people usually need a certain exercise volume to maximize exercise benefits. Therefore, we developed a wall squat program using a larger knee ankle system to arrange more sets than those in wall squat procedures in previous studies [38,39].

The wall squat exercise required the participants to stand with their back against the wall, feet parallel at shoulder width, tibia vertical. A goniometer was used to guide the subjects until they reached a knee joint angle of 120 degrees. During the exercise, a small stool was placed below the subjects’ hips to prevent falling due to exhaustion.

### 2.5. Procedure of the Experiment

Before the experiment, the participants met in the laboratory and were taught about the requirements for wall squat and the experimental procedures. All the participants finished two exercises under two experimental settings (treatment and control) in order to learn how to immediately respond to the research staff’s instructions during the test. The experiment was taken during nine days between 2:00 p.m. and 5:00 p.m. Prior to testing, the participants were asked to maintain abstinence from food for 2 h, energy drink, caffeine, and alcohol for 24 h, and sports for 24 h [1]. On each day, one participant from group A and another from group B arrived at the test room and took a 10-min rest, then they followed the tests alternately. Each participant underwent four consecutive tests, and each test contained three periods (P1, P2, and P3) (Figure 3). During P1, the subjects were seated rest indices were measured for 2 min (the video was not played in both experimental settings during this period). During P2, the subjects started wall squat and watched the nature video or black screen for 2 min, then they were asked to report the perceived exertion and affective responses at the end of P2. During P3, the subjects were seated again to take a 2-min rest. There was a 3-min interval between the tests for research staff to prepare the video or black screen and provide subjects with an extra rest.

### 2.6. Testing Environment

The experiment was carried out in a 10 m × 10 m strength training room, and redundant equipment was removed. A desk was placed on which to set the monitor, and a stool was placed in front of the table and against the wall for resting and testing. A window was covered by a shade curtain, and indoor illumination came from LED lighting. The ambient temperature was kept at around 20 °C, humidity at around 60%, and noise below 40 dB.

### 2.7. Outcome Measurements

Heart rate (HR) was monitored using a Polar H10 chest belt (Polar, Kempele, Finland), and the measurement for HR lasted three periods. The recorded R–R intervals were processed via Kubios HRV software (version 3.4.3, Kuopio, Finland). The HRV was assessed based on the entire period (2 min), including the fast and slow variation phases of HR during and after exercise (P2 and P3), in line with the previous HRV measurement for wall squat exercise [33]. As we used an ultra-short-term measurement (<5 min), several HRV indices were selected that are valid for the 2-min measurement, including the standard deviation of normal-to-normal RR intervals (SDNN), the root mean square of successive differences (RMSSD), and the standard deviations of the Poincaré plot (SD1) [40,41]. These HRV indices have also been previously applied in ultra-short-term analysis for exercise, including isometric exercise [41,42]. Heart rate recovery (HRR) is an important index for cardiac adaption of exercise. In the current study, HRR was calculated as the change of peak HR at the end of exercise and the HR observed after 30 s (HRR30s) and 60 s (HRR60s), respectively [43]. 

Perceived exertion was measured at the end of the exercise period via the 10-point Rating of Perceived Exertion (RPE) scale, with 0 as no exertion and 10 as maximal exertion. Affective responses were measured using the Feeling Scale, an 11-point bipolar scale ranging from −5 to +5, which is suitable for quantifying affective response (between displeasure and pleasure) during an exercise [44]. Both perceived exertion and affective response were measured at the end of the wall squat exercise. 

### 2.8. Statistical Analysis

The distribution of data was tested using the Shapiro-Wilk normality test. Since the data were mainly non-normally distributed, a general linear mixed model (GLMM) was performed at individual periods to check the differences in measured indices between the treatment and control settings. The experimental setting (two levels: treatment and control) was entered as the fixed factor while the subjects (*N* = 18) were entered as the random factor. All the analyses were conducted in the SPSS software (version 25.0, SPSS Inc., Chicago, IL, USA). A *p*-value adjusted using Bonferroni correction of <0.05 was considered statistically significant.

## 3. Results

Indices including HR, SDNN, RMSSD, and SD1 were measured before the exercise to check the differences between the treatment and control settings at the baseline. The GLMM did not reveal a statistically significant difference in measurement indices between the two experimental settings (Figure 4).

In terms of the exercise period, statistically significant lower HR was observed in the treatment setting (median of 89.60 and an interquartile (IQR) of 20.94) than that in the control setting (median = 92.30, IQR = 24.95, *p* = 0.01) (Figure 5a). On the other hand, though higher median values of SDNN, RMSSD, and SD1 were observed in the treatment setting, the differences were not statistically significant (*p* > 0.05). Besides, statistically significantly lower values of perceived exertion (median = 6.00, IQR = 2.00) and higher values of affective responses (median = 1.00, IQR = 2.00) were observed in the treatment setting compared to those in the control setting (median = 6.75 and 1.00, IQR = 2.38 and 1.00, *p* = 0.021 and 0.017, respectively) (Figure 5e,f). 

During the recovery period, statistically significantly lower value of HR was observed in the treatment setting (median = 84.18, IQR = 16.58) than that in the control setting (median = 86.30, IQR = 16.28, *p* = 0.001) (Figure 6a). Meanwhile, statistically significantly higher values of RMSSD (median = 34.88, IQR = 24.52) and SD1 (median = 24.75, IQR = 17.41) were observed in the treatment setting than those in the control setting (median = 31.56 and 22.39, IQR = 16.73 and 11.89, *p* = 0.004 and 0.003, respectively) (Figure 6c,d), but no significant difference was found in values of SDNN between the two experimental settings (*p* = 0.412, Figure 6b). Regarding the HRR, no statistically significant difference was found in HRR30s or HRR60s between the experimental settings (*p* > 0.05, Figure 6e,f).

## 4. Discussion

In the current study, we found that video viewing induced a lower value of perceived exertion, which supports our first hypothesis that nature video can reduce perceived exertion during wall squat exercise. These results are similar to those of Akers et al. [20], who found that cycling while watching a video of the rural environment decreased the perception of exertion. In their study, participants were asked to maintain a constant moderate intensity of cycling exercise, which minimized the impacts of varied exercise intensities. Similarly, exercise intensity of the wall squat program was decided by knee joint angle [1], which may further underline the effect of nature video viewing alone. However, the current study and that of Akers et al. [20] only recruited males, therefore the finding on the decreased perception of exertion is limited to males. Regarding the affective response, higher values were observed in the control setting, indicating greater pleasure under nature video viewing, which supports our first hypothesis. The effect we observed was similar to that of music on physical activity [45,46], which indicated the positive role of audiovisual intervention in exercise psychology. According to previous studies, nature-based stimuli including sights and sounds of nature have the potential to reduce negative psychological outcomes such as stress and pain [21]. Such positive effects might be achieved via the distraction effect of the nature feature [47]. Besides, due to environmental education and propaganda, the role of nature in health improvement has been underlined in public awareness [48,49]. As blinding is hard to carry out for the interventions used in the current study [17], a placebo effect may contribute to positive psychophysiological changes during exercise, although this may be regarded as only a part of nature-based interventions [50].

Theoretically, cardiac responses are affected by psychophysical or environmental stresses, and autonomic response to exercise leads to pronounced tachycardiac response with a consequent drop in HRV [35,51]. In the current study, we observed a statistically significantly lower HR for video viewing than in the counterpart, which partly supports our second hypothesis that nature video viewing may buffer HR growth during exercise. These results are consistent with those of Briki and Majed [52], who reported that walking in a simulated green environment induced a significantly lower HR when compared to walking in red and white conditions. The study identified the relaxing effect of the color green on the human organism. In terms of HRV, though higher values of SDNN, RMSSD, and SD1 were observed for nature video viewing during exercise, the differences between the two experimental settings were not statistically significant, which is not consistent with our findings on HR, and does not support our second hypothesis. According to previous studies, this inconsistency might result from different interpretations of the two cardiac indices, as HR is a result of both parasympathetic and sympathetic activity [53], while HRV may be predominantly determined by parasympathetic or sympathetic activity at different states [54,55]. In the published literature, walking in natural environments was reported to improve emotional states, thus inducing higher HRV than walking in non-natural environments [21,22]. These walking programs were designed as self-paced walking. By comparison, though our wall squat exercise induced a similar final HR (<100 bpm) to those found in the forest walking studies, the isometric exercise is not self-adjusted, which might induce higher mental stress during exercise and reduce the environmental effects. Besides, participants in the above two studies were all or mainly women, which may result in different sensibilities to nature-based interventions. However, whether gender difference can affect the effectiveness of nature-based interventions is still unknown, and needs further investigation.

In the recovery period, statistically significantly higher values of RMSSD and SD1 were observed for nature video viewing, and so were the values of HR. These results are similar to those found by Gladwell, et al. [56], who reported that nature picture viewing increased subjects’ RMSSD values. As RMSSD and SD1 were the two key indices for cardiac autonomic recovery after exercise [35,43], these results partly support our third hypothesis that nature video viewing may improve cardiac autonomic recovery after exercise [32]. This positive impact may be due to the physiological relaxation effect of nature, which is reported to enhance parasympathetic activity and suppress sympathetic activity [22]. However, in terms of HRR, no statistically significant difference was observed between the two experimental settings on the basis of 30s and 60s recovery, which does not support our third hypothesis. In a relevant study, Elisabet, et al. [57] also observed a non-significant difference in HRR among three natural and one unnatural environments. According to previous studies, HRR is associated with exercise characteristics [35], which may receive less impact from psychological factors, and this may partly explain the ineffectiveness of the nature video on HRR.

In general, our results indicate that nature-based video may improve exercise experience, similar to the known benefits of exposure to real nature environments [22,58,59]. However, there is an obvious difference between the current and previous studies. Due to the characteristics of the wall squat exercise, only a short-term exposure to nature-based digital stimuli was conducted in the current study. By comparison, the previous nature exposure usually lasted for dozens of minutes. Empirically, the effect of nature exposure is affected by the dose of exposure, and duration is an important dose-related factor [60,61,62]. Therefore, the effectiveness of nature-based digital stimuli may benefit from a longer exposure. However, the dose of exposure to nature-based digital stimuli has not yet been investigated, and could be different to exposure to real nature, due to the display devices. Nevertheless, our findings provide evidence for the effectiveness of nature-based digital stimuli in green exercise. 

However, there are some limitations to the current study. First, we only recruited young men from the university due to our limited experimental conditions, so the psychophysiological outcomes of women are still unclear, which needs further investigation. Besides, although offset by perhaps offset by the self-control design, the habits of participants such as drinking and smoking may affect performance in exercise testing and response to nature. Further work might consider extra inclusion criteria. Second, exercise ability such as static muscle endurance can affect the response to exercise, including fatigue and mental pressure, which may affect the effectiveness of nature-based interventions. Further study may focus on subjects of more diversity and a larger sample size to produce more reliable results. Third, the effectiveness of nature-based interventions may vary with replications, and whether the sense of novelty played a role in interventions is still unknown, which may be a common problem for single nature-based studies, so further studies using multi-stage interventions are needed. Finally, apart from the above points, the effectiveness of nature video itself on cardiac responses still needs clarification, particular for the resting indices, because we only displayed the nature video temporarily in the exercise and recovery period for the wall squat exercise, and the sensibility to nature-based interventions may be affected by individuals’ physiological states [63]. Besides, the effects of viewing nature images or videos on cardiac responses may vary with viewing duration and display device [8]. Therefore, future studies may resort to advanced devices and use different designs to find out an effective way to deliver nature benefits via digital media.

## 5. Conclusions

The presented study explored a new form of green exercise and verified its effects on perceived exertion, HR, and HRV indices via nature video. Our results demonstrated that nature video viewing induced higher pleasure, lower perceived exertion and HR during wall squat exercise, and enhanced cardiac autonomic recovery after the exercise. However, due to the limited sample size of men only, further studies using multiple devices and diverse participants are needed to provide a practical indication for the general population.

## Figures and Tables

**Figure 1 ijerph-18-05554-f001:**
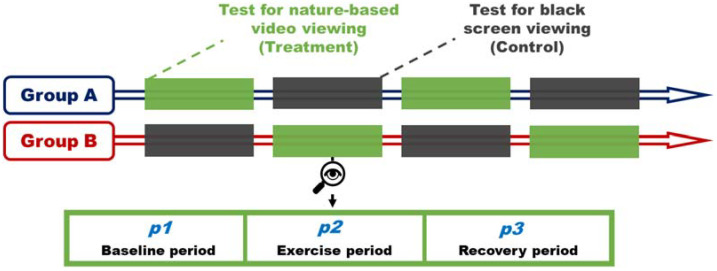
Diagram of study design.

**Figure 2 ijerph-18-05554-f002:**
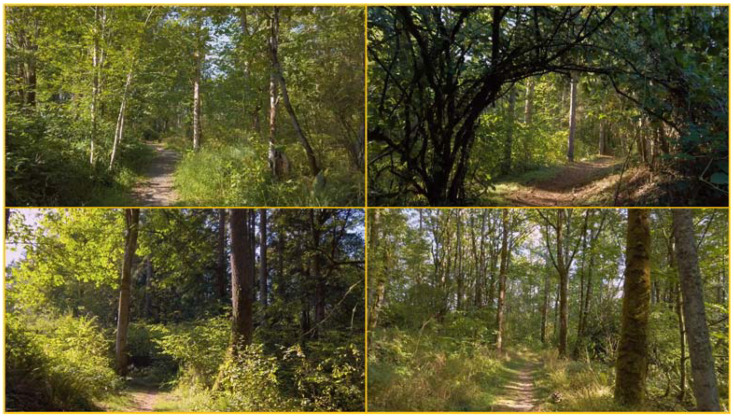
View of the nature video.

**Figure 3 ijerph-18-05554-f003:**
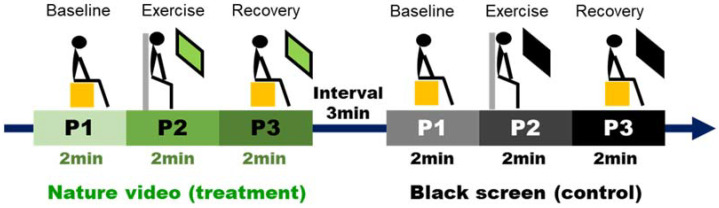
Procedure of the experiment.

**Figure 4 ijerph-18-05554-f004:**
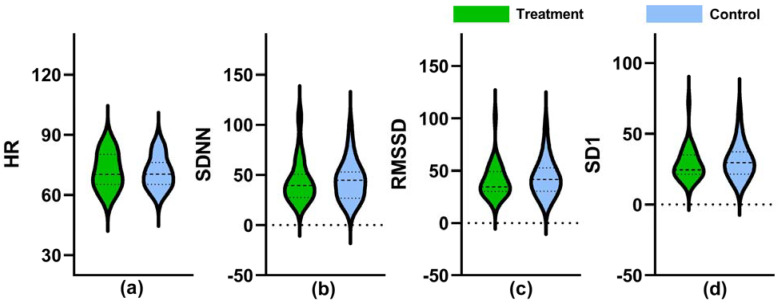
Values of HR (**a**), SDNN (**b**), RMSSD (**c**), and SD1 (**d**) recorded in the baseline period. The dashed lines in the violin plots indicate median and quartile values.

**Figure 5 ijerph-18-05554-f005:**
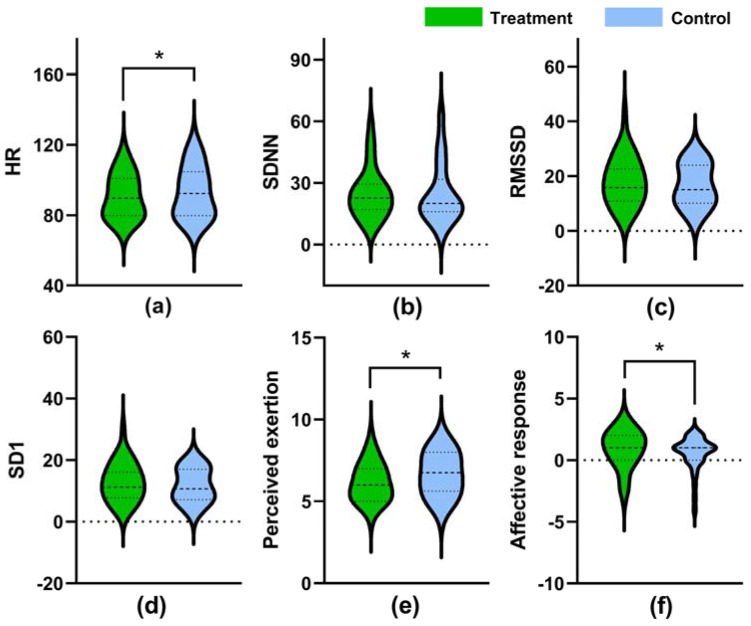
Values of HR (**a**), SDNN (**b**), RMSSD (**c**), SD1 (**d**), perceived exertion (**e**), and Affective response (**f**) recorded in the exercise period. The dashed lines in the violin plots indicate median and quartile values. *: *p* < 0.05 (treatment vs. control), adjusted by Bonferroni correction.

**Figure 6 ijerph-18-05554-f006:**
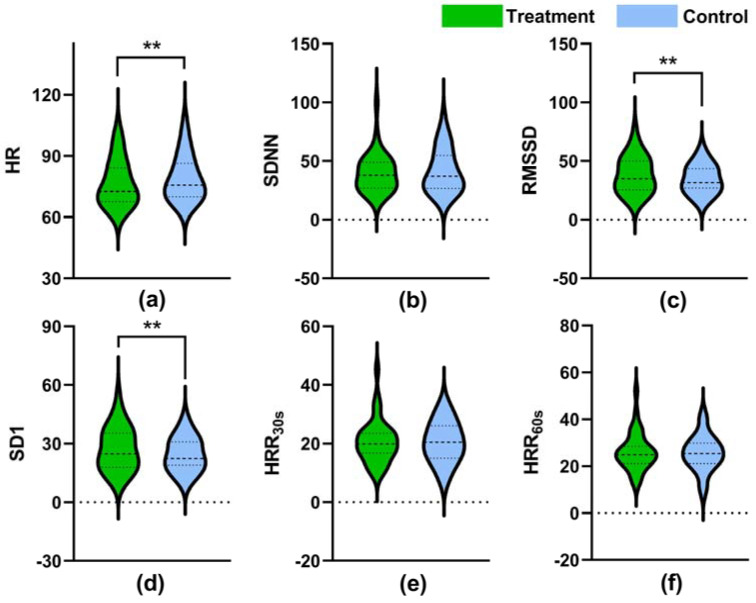
Values of HR (**a**), SDNN (**b**), RMSSD (**c**), SD1 (**d**), HRR30s (**e**), and HRR60s (**f**) recorded in the recovery period. The dashed lines in the violin plots indicate median and quartile values. **: *p* < 0.01 (treatment vs. control), adjusted by Bonferroni correction.

**Table 1 ijerph-18-05554-t001:** Characteristics of the participants recruited in the study.

Gender	*N*	Age (year)	Height (cm)	Weight (kg)	BMI (kg/m^2^)
Male	18	27.94 (6.13)	177.33 (5.73)	73.72 (9.28)	23.36 (1.90)

values present mean (SD).

## Data Availability

The data presented in this study are available on request from the corresponding author. The data are not publicly available due to privacy.

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
