# Peer review of "Green Exercise: Can Nature Video Benefit Isometric Exercise?"

_ijerph, 2021, doi:10.3390/ijerph18115554_

Round 1

Reviewer 1 Report

I still have the same methodological issues as in the first version. Yet also, my main concern is that i dont think it has the quality, soudness  and significance of content to be published in this journal, thus i suggest and encourage the authors to submit elsewhere. 

Furthermore, i will like to point out the aferomentioned comment of reviwers: 

- it is unclear if the ‘physical effort’ put in by the subjects in the isometric exercise was equivalent between the NATURE and BLACK trials.

- were the subjects informed of the purpose of the study? If they were, there is a possibility of a placebo and nocebo effects within the NATURE and BLACK trials, respectively.

- is it possibly that the ‘benefits’ of the NATURE trial is due to the ‘novelty’ or first exposure effects, rather than the exposure to NATURE perse? This should be discussed in the study.

. furthermore, In respect of the design of the “isometric exercise”, it is known that it normally takes a few minutes to reach a steady state (e.g. HR) in a submaximal (dynamic) exercise, even after a proper warm-up. In this study, there was no warm-up, and the total exercise time was only 2-min.

The authors may need to provide some background or cite relevant literature for the possible HR changes in such a protocol and discuss whether this would have affected the key outcome measures (HR and HRV).

Reviewer 2 Report

None.

Reviewer 3 Report

The authors have properly addressed most of my comments. Thank for their efforts.

There are still a few minor issues in the revised manuscript that could be addressed.

Line 79: consider replace “process” with “response”

Line 86: consider to define further for “moderate”. Is it about the intensity or duration or what?

I am not convinced that the wall squat can be considered similar to walking, as one is static and the other is dynamic, that will have different physiological effects on cardiovascular responses.

Line 125-126: Normally standard deviation should be presented in this type of investigation. Also, did you measure these variables accurate to two decimal positions?

Line 140: why “lower intensity” is used here. How does this relate to the “moderate” mentioned above?

Line 158-159: thank the authors to clarify this point. But my previous comment was about whether to watch the video during rest would affect HR and HRV. In this current research design, the exercise+video  vs exercise+blank would control for the effect of visual stimulation during exercise, but it is still unknow whether to watch video itself would affect the HR (that affect the baseline). Could this be another limitation of the study?

Author Response

This manuscript is a resubmission of an earlier submission. The following is a list of the peer review reports and author responses from that submission.

Round 1

Reviewer 1 Report

The study design and statistical treatment appear to be correct. Given the variables to be measured, it would be interesting for the authors to detail more about the inclusion and exclusion criteria, since in the case of the consumption of coffee or other ergogenic aid prior to or as a habitual consumer, this would affect the result of the study.

Reviewer 2 Report

The work presented in this manuscript examined the effect of watching a video of forest walking during a 2-min indoor static wall squat on HR and HRV, during a 2-min recovery on HR, in a group healthy male participants. It was concluded that the nature video viewing induced higher pleasure, lower perceived exertion and HR during the all squat, and enhanced cardiac autonomic recovery after the exercise.

It is an interesting study with potentially new contributions to the area of knowledge in respect of investigating the psycho-physiological effect of a specific visual stimulation during an isometric exercise, if the authors could clarify several areas in the manuscript. I offer the following critiques for authors’ consideration aiming to improve the quality of the manuscript.

I would expect a rationale in the Introduction for analyzing HRV in this study. The readers would appreciate some (physiological) background information about HRV in relation to the aims of this study, and specifically in relation to isometric exercise.

In respect of the design of the “isometric exercise”, it is known that it normally takes a few minutes to reach a steady state (e.g. HR) in a submaximal (dynamic) exercise, even after a proper warm-up. In this study, there was no warm-up, and the total exercise time was only 2-min. The authors may need to provide some background or cite relevant literature for the possible HR changes in such a protocol and discuss whether this would have affected the key outcome measures (HR and HRV).

Further, there was no information about the participants’ fitness and normal physical activity level (e.g. at least the maximal tolerance time for the wall squat), that might have influenced the HR response. Could this be a limitation of the study.

For the video of forest walking, was the video a long, continuous clip (e.g. more than 4 min of the experimental time), or a shorter clip (e.g. 1 or 2 min) that was shown cyclically to the participants?

Would the video watching affect resting HR, e.g. during the baseline period? If so, that might have affected the results. Could this be another limitation if it was not included in the design?

In the Figures, please define what the horizontal dash lines represent in the bubbles.

In the Conclusions, the meaning of “cardiac process” is not clear. Please be more specific, e.g. HR or some indices of HRV.

In the Abstract, please spell out the abbreviations. Also, in the last statement “…might improve the efficacy of isometric exercise…”, the authors might need re-consider what “efficacy” means here, e.g. impact on exercise time, or physiological benefits. It is possibly better to say “that requires further investigation”.

Reviewer 3 Report

Green Exercise: Can Nature Video Benefit Isometric Exercise?

This is a concise, well-written piece of article on a simple experimental trial.

Major issues:

- it is unclear if the ‘physical effort’ put in by the subjects in the isometric exercise was equivalent between the NATURE and BLACK trials.

- were the subjects informed of the purpose of the study? If they were, there is a possibility of a placebo and nocebo effects within the NATURE and BLACK trials, respectively.

- is it possibly that the ‘benefits’ of the NATURE trial is due to the ‘novelty’ or first exposure effects, rather than the exposure to NATURE perse? This should be discussed in the study.

Minor issues

- Line 117. No mentioned of the “type’ of sounds that were player during the NATURE and BLACK trials

- Line 127. What were the environmental conditions of the experimental trials – e.g., what were the humidity and ambient temperature values. What were the surrounding conditions of the trials – were the exercise conducted in empty, walled-up room with no windows to the outside view – because if there were windows or other personnel in the room – these will act as unnecessary distractions and/or interventions to the trials.

- Line 270. What is ‘indic’?